# An Adaptive Fusion Algorithm for Depth Completion

**DOI:** 10.3390/s22124603

**Published:** 2022-06-18

**Authors:** Long Chen, Qing Li

**Affiliations:** 1Institute of Microelectronics, Chinese Academy of Sciences, Beijing 100029, China; 2University of Chinese Academy of Sciences, Beijing 100049, China

**Keywords:** depth completion, depth estimation, adaptive mechanism, multi-modal fusion, convolutional neural networks

## Abstract

Dense depth perception is critical for many applications. However, LiDAR sensors can only provide sparse depth measurements. Therefore, completing the sparse LiDAR data becomes an important task. Due to the rich textural information of RGB images, researchers commonly use synchronized RGB images to guide this depth completion. However, most existing depth completion methods simply fuse LiDAR information with RGB image information through feature concatenation or element-wise addition. In view of this, this paper proposes a method to adaptively fuse the information from these two sensors by generating different convolutional kernels according to the content and positions of the feature vectors. Specifically, we divided the features into different blocks and utilized an attention network to generate a different kernel weight for each block. These kernels were then applied to fuse the multi-modal features. Using the KITTI depth completion dataset, our method outperformed the state-of-the-art FCFR-Net method by 0.01 for the inverse mean absolute error (iMAE) metric. Furthermore, our method achieved a good balance of runtime and accuracy, which would make our method more suitable for some real-time applications.

## 1. Introduction

Accurate depth information is essential for many computer vision applications, such as autonomous driving, SLAM (simultaneous localization and mapping) and drones. Most active depth sensors can easily acquire depth information for indoor scenes. In outdoor scenes, depth perception mainly relies on stereo vision or LiDAR (light detection and ranging) sensors. However, stereo vision methods tend to have lower accuracy for long-distance regions and LiDAR sensors can only obtain sparse depth information. Depth completion has received more and more attention as a solution to this problem.

Depth completion methods can generally be divided into two categories: depth-only [1,2,3] and image-guided methods [4,5,6,7,8,9]. Depth-only methods generate dense images directly from sparse inputs. Uhrig et al. [10] first proposed sparse invariant convolution to handle sparse inputs more effectively. On this basis, Huang et al. [2] proposed a hierarchical multi-scale sparse invariant network.

Although depth completion that does not use RGB images for guidance can achieve good results, the error rate is still large. Due to the rich textural information of RGB images, researchers have started to use RGB images to guide depth completion. Therefore, it is very important to effectively integrate the information from RGB images with the information from LiDAR sensors [7].

Older methods directly inputted RGB images and LiDAR data into a convolutional neural network (CNN) and did not perform well (early fusion) [6]. Some recent methods take RGB images and LiDAR data as the inputs of convolutional neural networks separately and then fuse the features that are extracted by the convolutional neural network (late fusion) [7,8]. These methods produce significant improvements over the earlier methods. For example, Qiu et al. [4] used “depth surface normals” that were obtained from RGB images to guide depth completion. While this study achieved great results, the authors basically used feature concatenation or element-wise addition for the feature fusion. Since RGB images and LiDAR data come from different sensors, simple feature concatenation or element-wise addition does not yield the best results [5,6]. Accordingly, Tang et al. [7] trained a guided convolutional neural network to fuse the features from RGB images with those from sparse LiDAR data. This method achieved good results when using the KITTI depth completion dataset.

To sum up, effectively fusing the features of RGB images with those of sparse LiDAR data is very important. To this end, this paper proposes an adaptive fusion mechanism. Unlike traditional feature concatenation or element-wise addition, our method divides the feature maps from RGB images and LiDAR sensors into different parts and utilizes an attention mechanism to generate a different kernel weight for each part. In this way, our network generates content-dependent kernels for multi-modal feature fusion. This makes our method more advantageous during inference. Our method achieved good results when using the KITTI and NYUv2 datasets. The experimental results showed that our algorithm achieved a good balance between runtime and accuracy and outperformed the traditional algorithms in terms of robustness.

The structure of this paper is organized as follows. We discuss related work in Section 2. Section 3 introduces our method. Finally, we evaluate the results of our algorithm from all tasks quantitatively and qualitatively in Section 4.

## 2. Related Work

The previous methods can be roughly divided into two categories, depending on whether there are RGB images to guide the depth completion: image-guided methods and depth-only methods. We briefly review these methods in this section.

### 2.1. Depth-Only Methods

When accurate but low-resolution depth sensors were widely used, researchers began to focus on the problem of turning sparse depth into dense depth. Some early methods utilized compressed sensing theory [1] or wavelet analysis [3] to solve this problem. In recent years, with the development of deep learning and especially the development of convolutional neural networks, deep learning-based depth completion methods have achieved good results. Chodosh et al. [11] used combined compressed sensing to handle sparsity while filtering out missing values using binary masks. Uhrig et al. [10] proposed a sparse invariant convolutional neural network to handle sparse data or features by observing masks. Huang et al. [2] extended the concept of sparse invariant convolution to other operations (summation, upsampling and concatenation) to implement multi-scale neural networks. Eldesokey et al. [12] replaced the binary mask with continuous confidence in sparse invariant convolution. Although depth-only methods have achieved great results, the error rate is still very large. Therefore, researchers have begun to focus on methods that are guided by RGB images.

### 2.2. Image-Guided Methods

Due to the poor performance of the depth-only methods, methods that use RGB images as guides were proposed [13,14,15]. Schneider et al. [16] proposed a method to supplement the sparse depth map with intensity hints and object boundary hints. Although they achieved certain results, they did not consider how to fuse the image information. Ma et al. [5] used an early fusion scheme combining sparse depth inputs with the corresponding RGB images and experimentally demonstrated that the scheme could perform well. Zhang and Funkhouser [17] achieved promising results using neural networks to predict the dense surface normals and occlusion boundaries from RGB images as auxiliary information to supplement sparse depth data. Based on this, Qiu et al. [4] extended the generated surface normals into intermediate representations of outdoor datasets to guide depth completion. Jaritz et al. [8] used neural networks with large receptive fields to process sparse LiDAR data and used variable sparse depth maps to train the network. Hui et al. [18] proposed a convolutional neural network incorporating RGB image-guided information from different stages and trained it using high-frequency regions. Li et al. [13] then proposed a multi-scale guided cascaded hourglass network for depth completion.

Currently, the top ranked methods on KITTI’s official leaderboard [7,19,20] are designed mostly to better integrate the information from RGB images. For example, Tang et al. [7] designed a guided network to predict kernel weights from RGB images. These predicted kernels can then be applied to extract deep image features.

Therefore, in order to better utilize the image information, we designed an adaptive fusion mechanism to fuse information from different sensors. Our method divides the feature maps of RGB images and LiDAR sensors into different parts and utilizes an attention mechanism to generate content-dependent kernels for multi-modal feature fusion.

## 3. Methods

We considered feature $S\in R^{H\times W\times C}$ (where *H*, *W* and *C* are the height, width and number of channels in the input feature, respectively), which was obtained by processing LiDAR data through a convolutional neural network, and feature $I\in R^{H\times W\times C}$, which was generated by the RGB images. Depth completion aimed to fuse $S\in R^{H\times W\times C}$ and $I\in R^{H\times W\times C}$ to generate the new feature $D\in R^{H\times W\times C}$. The goal of our method was to generate kernel weights based on the feature content and use these kernels to fuse the multi-modal features. In this section, we first briefly review the traditional feature concatenation method and its limitations. We then describe the proposed adaptive fusion mechanism.

### 3.1. Motivation

We first briefly introduce the traditional feature concatenation method. For the feature $\left[S,I\right]\in R^{H\times W\times C}$, the feature concatenation method uses convolutional operations to generate $D\in R^{H\times W\times C}$. We assumed that the convolution in the feature fusion method had no padding and that the stride was 1. The feature concatenation method could then be described by the following formula: (1)$$D_{:,:,i}=\sum_{i=1}^{C}{\left[I,S\right]}_{:,:,i}⊗K_{:,:,i}^{\left(j\right)}$$
where ⊗ denotes the convolutional operation.

Since $S\in R^{H\times W\times C}$ and $I\in R^{H\times W\times C}$ were from different sensors, they did not fuse well using the feature concatenation method [4,5]. For such multi-modal data fusion, we needed:Fusion using a different kernel for each pixel location;The kernel to be automatically generated according to the content.

From these two points, we proposed an adaptive fusion mechanism.

### 3.2. Adaptive Fusion Mechanism

For each pixel position, when different convolutional kernels are used to aggregate the information, it is easy to calculate the parameter size of the convolutional kernel as H×W×k×k×C, which produces a huge number of parameters. To solve this problem, as shown in Figure 1a, we divided the features into n × n parts. In the experiments, we took the size of n as 3 and the feature map of each part was $\left[S,I\right]\in R^{H/3\times W/3\times C}$. For each part, when a kernel of size 1 was used to aggregate the information from each pixel location, the number of parameters was reduced to 3×3×k×k×C and our method could be expressed as: (2)$$D^{H/3\times W/3\times C}={\left[I,S\right]}^{H/3\times W/3\times C}⊗K^{G}$$
(3)$$D^{H\times W\times C}=\overset{9}{\underset{i=1}{⋃}}D_{i}^{H/3\times W/3\times C}$$
where $K^{G}$ is the content-dependent convolutional kernel, ⊗ is the convolutional operation and ⋃ is the union operation.

Figure 1a shows the overall design of our method. Figure 1b shows our “adaptive kernel generation” (AKG) module. For the feature $\left[S,I\right]\in R^{H/3\times W/3\times C}$, we first used global average pooling to compress the information to obtain $K_{1}^{C}$, which could then be expressed as: (4)$$K_{1}^{C}=AvgPool\left({\left[I,S\right]}^{H/3\times W/3\times C}\right)$$

Then, we used the attention mechanism to capture the correlation between the channels. The formula could be expressed as: (5)$$A\left(K_{1}^{C}\right)=\sigma \left(W_{1}⊗\left(\mathrm{ReLU}\left(W_{0}⊗K_{1}^{C}\right)\right)\right)\times K_{1}^{C}+K_{1}^{C}$$
(6)$$K_{2}^{M}=\sigma \left(W_{2}⊗\left(A\left(K_{1}^{C}\right)\right)\right)$$
where σ is the sigmoid function, $W_{0}$, $W_{1}$ and $W_{2}$ are convolutional kernels of size 1 and ⊗ represents the convolutional operation.

Then, we performed a matrix multiplication of $K_{2}^{M}$ and $K^{M\times C_{i}\times C_{o}\times 1\times 1}$ to obtain $K^{G}$:(7)$$K^{G}=K_{2}^{M}⊙K^{M\times C_{i}\times C_{o}\times 1\times 1}$$
where ⊙ represents the matrix multiplication and the value of $K^{M\times C_{i}\times C_{o}\times 1\times 1}$ is calculated by back-propagation. In this specific experiment, we set *M* to 2.

**Network architecture**. Our method can be used as an independent module, so we validated our method using two classical network architectures [21,22] (note that the techniques that were developed in this work do not involve new network architectures). The two network architectures are shown in Figure 2. The network architecture in Figure 2a is similar to HR-net [21] and was used to extract the features from RGB images (Net1) and the network architecture in Figure 2b is similar to U-net [22] and GuidedNet [7] (Net2).

**Computational efficiency analysis**. In contrast to the feature concatenation method, our method consists of two parts: kernel generation and feature fusion. The time complexity of Formula (1) was $O\left(C_{\mathrm{in}\;}\times C_{\mathrm{out}\;}\times K^{2}\times H\times W\right)$ and the time complexity of Formulae (2) and (3) was $O\left(C_{\mathrm{in}\;}\times C_{\mathrm{out}\;}\times K^{2}\times H\times W\right)$. For the adaptive kernel generation, the time complexity was $O\left((C_{in}\times C_{out}+C_{in}\times M)\times n\times n\right)$. We assumed that the running times of the proposed method and the feature concatenation method were $T_{o}$ and $T_{c}$, respectively. Then:(8)$$\begin{array}{cc} \frac{T_{o}}{T_{c}} & =\frac{O\left(C_{\mathrm{in}\;}\times C_{\mathrm{out}\;}\times K^{2}\times H\times W\right)+O\left((C_{in}\times C_{out}+C_{in}\times M)\times n\times n\right)}{O\left(C_{\mathrm{in}\;}\times C_{\mathrm{out}\;}\times K^{2}\times H\times W\right)}\\ \; & \approx 1+\frac{n\times n}{K^{2}\times H\times W} \end{array}$$

Therefore, the increased computation time of our method was negligible compared to that of the feature concatenation method.

## 4. Experiments

In the experiments, we used an ADAM [23] optimizer with an initial learning rate of 0.001. Every 30,000 iterations, the learning rate was reduced by half. We used 1 RTX 3090 GPU for batch training, we set the batch size to 8 and used the root mean square error (RMSE) as the loss function. In this section, we first introduce our dataset and evaluation metrics. We then present our ablation experiments and the quantitative and qualitative comparisons to state-of-the-art methods. Finally, our robustness testing experiments are presented.

### 4.1. Datasets

**KITTI.** The KITTI depth completion dataset [24] that we used contains 85,898 training images, 1000 testing images and 1000 validation images. KITTI provides an official public leaderboard and the results of the testing dataset can only be tested after being submitted to the official website. The real depth data in the KITTI dataset was obtained by LiDAR scanning and the upper part of the images do not contain sparse depth measurements. Thus, we followed the same settings as those in [7] by bottom cropping the input image to 1216 × 256 for training and inference (the original size of the image was 1216 × 375).

**NYUv2.** The NYUv2 dataset [25] consists of RGB and depth images that were captured by a Microsoft Kinect in 464 indoor scenes. Following the previous depth completion method [7], we trained our network using the training dataset and evaluated it using the 654 officially labeled testing set.

### 4.2. Evaluation Indicators

According to KITTI’s official benchmark [24], all algorithms need to be evaluated based on the following metrics: root mean square error (RMSE); mean absolute error (MAE); inverse root mean square error (iRMSE); and inverse mean absolute error (iMAE).
(9)$$\mathrm{RMSE}={\left(\frac{1}{\left|D\right|}\sum_{u,v\in D}{\left|D^{\mathrm{gt}}\left(u,v\right)-D\left(u,v\right)\right|}^{2}\right)}^{0.5}$$
(10)$$\mathrm{MAE}=\frac{1}{\left|D\right|}\sum_{u,v\in D}\left|D^{\mathrm{gt}}\left(u,v\right)-D\left(u,v\right)\right|$$
(11)$$\mathrm{iRMSE}={\left(\frac{1}{\left|D\right|}\sum_{u,v\in D}{\left|\frac{1}{D^{\mathrm{gt}}\left(u,v\right)}-\frac{1}{D(u,v)}\right|}^{2}\right)}^{0.5}$$
(12)$$\mathrm{iMAE}=\frac{1}{\left|D\right|}\sum_{u,v\in D}\left|\frac{1}{D^{\mathrm{gt}}\left(u,v\right)}-\frac{1}{D(u,v)}\right|$$
where *D* represents the output, $D^{\mathrm{gt}}$ represents the true depth value and (u,v) represents the pixel coordinates.

### 4.3. Comparison to State-of-the-Art Methods

#### 4.3.1. Kitti Dataset

We compared our method (using the Net1 architecture) to state-of-the-art methods using the KITTI depth completion testing dataset. Table 1 presents the quantitative comparison of our method to the other top-ranking published methods on the KITTI leaderboard. In particular, our method comprehensively outperformed the DeepLiDAR method [4], which was trained with additional data.

Our method was also highly competitive compared to the NLSPN [32], GuidedNet [7] and FCFR-Net [15] methods. Specifically, compared to the FCFR-Net [15] method, our method reduced the iMAE metric by 0.01. Our method outperformed the GuidedNet method [7] on both the MAE and iMAE metrics and improved the runtime by about 0.09 s/frame. Compared to the NLSPN [32] method, our method reduced the RMSE index by 1.52 mm and was far superior to the NLSPN [32] method in terms of runtime. This is because the NLSPN [32] method employs non-local spatial iterations to optimize the final depth value, which is very time-consuming (NLSPN [32], GuidedNet [7] and our method were all tested using RTX2080ti).

Our method did not need much extra runtime but could improve the final performance. This would make the method in this paper more suitable for some applications that require high real-time performance. Figure 3 shows the qualitative comparison of our method to some of the other methods [12,27]. Our method preserved more details in the boundary regions and performed better on small objects.

#### 4.3.2. Nyuv2 Dataset

To test the performance of our method for indoor scenes, we trained and evaluated our network (using Net1 architecture) using the NYUv2 dataset [25]. Following the existing methods [7,27], we trained and evaluated our method using the settings of 200 and 500 sparse LiDAR samples (200 or 500 depth pixels were randomly sampled from a dense depth image and used as the inputs along with the corresponding RGB images). The quantitative comparison to the other methods is presented in Table 2. It can be seen that our method outperformed most methods.

### 4.4. Ablation Study and Analysis

To analyze the impact of our fusion method on the final performance, we conducted a comparative study using the KITTI validation dataset. We validated our method on two networks, the structures of which are shown in Figure 2. The following modules were compared: feature concatenation (Concat) and element-wise addition (Add).

In the experiments, we replaced the adaptive fusion module with the Add or Concat operations and kept the other network components unchanged. The performance comparison of different operations is shown in Table 3.

As can be seen from Table 3, using simple element-wise addition or concatenation made the final result significantly worse compared to using the adaptive fusion mechanism. Specifically in the Net1 architecture, our method reduced the RMSE metric by about 28 mm compared to the Concat operation and by about 41 mm compared to the add operation. It can also be seen that Concat worked better than the Add operation. This is because RGB images and depth data come from different sensors. This result further verified our previous analysis.

### 4.5. Robustness Test

Since depth data are acquired by LiDAR sensors, there are different degrees of noise in practical applications. Therefore, it is important to test the performance of depth completion algorithms under different noise levels. In this section, we used the Net1 architecture to test the robustness of our method and compare it to several other methods. The models used in this section were all trained on raw data and tested directly with noisy data.

We randomly selected 5 to 30 percent of the data and added different levels of Gaussian noise. These Gaussian noises could simulate LiDAR data in rain or snow. We tested the robustness of our method under different noise percentages and different noise intensities.

In the experiments, we set the mean to 0 and simulated different intensities of noise with different variances. The specific comparison data are shown in Figure 4c. We tested the robustness of the model by adding random noise and simulated LiDAR data for different harnesses by randomly discarding input depth values. The results are shown in Figure 4a,b.

Figure 4 shows that the robustness of our method outperformed all of the compared methods, demonstrating the superiority of our method.

## 5. Conclusions

This paper proposed a new fusion mechanism. Unlike previous methods, this mechanism generates convolutional weights based on location and content information to adaptively fuse information from different sensors. Extensive experiments verified the effectiveness of our adaptive fusion mechanism for depth completion. Our method not only showed strong performances for both indoor and outdoor scenes, but it also showed a strong generalization ability under different point densities and various lighting and weather conditions. Our method could also bring new inspiration to the field of image fusion [33]. Although our method achieved promising results, this paper did not explore the impacts of different network architectures on the final performance. We will try to design new network architectures in future work, as well as designing more elegant content-dependent kernels.

## Figures and Tables

**Figure 1 sensors-22-04603-f001:**
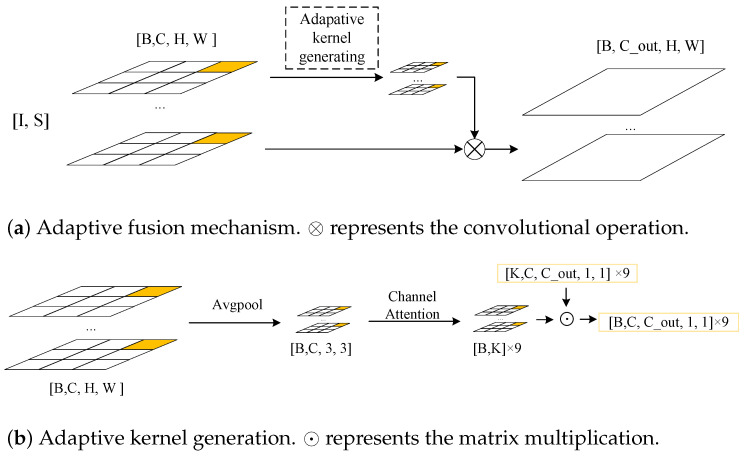
The adaptive fusion mechanism: (**a**) the overall flow of our adaptive fusion mechanism (for image feature *I* and depth map feature *S*, the adaptive kernel mechanism dynamically generates a kernel $K^{G}$ to fuse *I* and *S* and outputs the new depth feature D); (**b**) the adaptive kernel generation.

**Figure 2 sensors-22-04603-f002:**
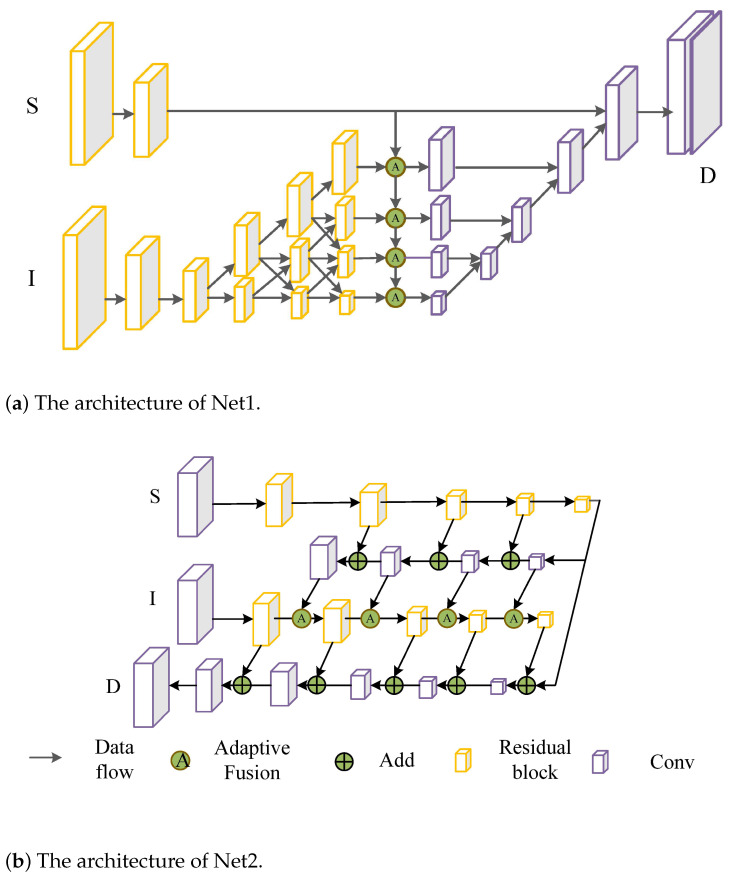
The network architectures: (**a**) the Net1 architecture; (**b**) the Net2 architecture.

**Figure 3 sensors-22-04603-f003:**
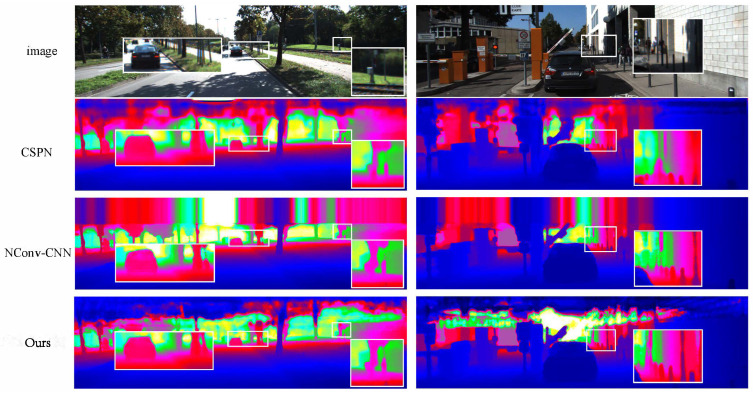
The qualitative comparisons were performed by CSPN [26] and NConv-CNN [12] using the KITTI testing dataset. The results are from the KITTI depth completion leaderboard, in which depth images are colorized along with the depth range. Our method (framed by the white boxes) achieved a better performance and recovered better details.

**Figure 4 sensors-22-04603-f004:**
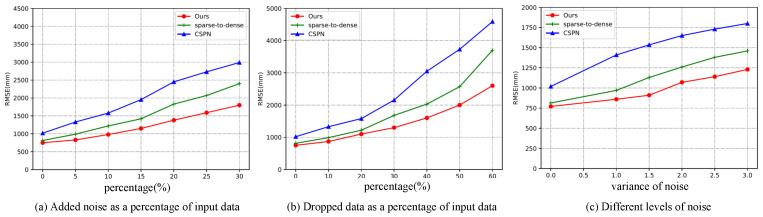
The robustness was tested using the validation dataset of the KITTI depth completion benchmark (Net1 architecture): (**a**) the randomly added noise as a percentage of the input data; (**b**) the randomly discarded data as a percentage of input data; (**c**) the addition of varying degrees of noise to 10 percent of the input data (0 means no noise was added).

**Table 1 sensors-22-04603-t001:** Performance comparison using the KITTI testing dataset. We tested our method using the Net1 architecture. The results were evaluated by the KITTI testing server and the different methods were ranked according to their RMSE. “Additional information” means that the algorithm was trained with additional data or labels. Note that most of the runtimes in the table are from their own papers and were tested on different GPUs, so only rough comparisons could be drawn.

Methods	Additional Information	RMSE	MAE	iRMSE	iMAE	Time (s)
DFuseNet [26]	×	1206.66	429.93	3.62	1.79	0.08
CSPN [27]	×	1019.64	279.46	2.93	1.15	1
HMS-Net [2]	×	937.48	258.48	2.93	1.14	-
Sparse-to-dense [5]	×	814.73	249.95	2.80	1.21	0.08
Cross-Guidanced [9]	×	807.42	253.98	2.73	1.33	0.2
PwP [28]	✓	777.05	235.17	2.42	1.13	0.1
DSPN [29]	×	766.74	220.36	2.47	1.03	0.34
DeepLiDAR [4]	✓	758.38	226.50	2.56	1.15	0.07
UberATG-FuseNet [30]	×	752.88	221.19	2.34	1.14	0.09
CSPN++ [31]	×	743.69	209.28	2.07	0.90	0.2
NLSPN [32]	×	741.68	**199.59**	**1.99**	**0.84**	0.20
**Ours**	×	740.16	215.69	2.28	0.97	**0.05**
GuidedNet [7]	×	736.24	218.83	2.25	0.99	0.14
FCFR-Net [15]	×	**735.81**	217.15	2.20	0.98	0.13

**Table 2 sensors-22-04603-t002:** Performance comparison using the NYUv2 dataset. The settings of both 200 samples and 500 samples were evaluated.

Samples	Methods	RMSE
	Sparse-to-dense [5]	0.204
	NConv-CNN [12]	0.129
500	CSPN [27]	0.117
	DeepLiDAR [4]	0.115
	**Ours**	0.104
	GuidedNet [7]	**0.101**
	Sparse-to-dense [5]	0.230
200	NConv-CNN [12]	0.173
	GuidedNet [7]	0.142
	**Ours**	**0.140**

**Table 3 sensors-22-04603-t003:** The results of the different fusion methods using the KITTI validation dataset.

Net Architecture	Fusion Methods	RMSE	MAE	iRMSE	iMAE
	Add	811.51	237.32	3.61	1.20
Net1	Concat	798.30	232.63	2.30	1.06
	**Ours**	**770.29**	**217.11**	**2.03**	**1.01**
	Add	812.59	239.48	3.67	1.24
Net2	Concat	803.77	233.67	2.49	1.08
	**Ours**	**778.29**	**222.41**	**2.21**	**1.03**

## Data Availability

No data is reported.

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
