# Peer review of "An Adaptive Fusion Algorithm for Depth Completion"

_sensors, 2022, doi:10.3390/s22124603_

Round 1
Reviewer 1 Report
“An adaptive fusion algorithm for depth completion” by Long Chen and Qing Li
Summary:
This article presents a novel technique for depth completion of LiDAR images through fusion with RGB images. The novelty is in the technique used for combining RGB and LiDAR data through adaptive convolution kernels based on the content and location of feature vectors. This new fusion technique provides similar results to the current leading state-of-the-art methods on the KITTI test set and improves on the indoor images in the NYUv2 dataset. When compared against pure addition or concatenation, the novel fusion technique performs better on the KITTI validation set. The technique also appears to be more robust to noise in the input data.
Review:
Major concerns
Lines 170 – 178: This speed comparison on other GPUs is not sufficient for analysis. Specific GPUs must be compared, as core frequency is not sufficient information to describe a GPU. The number of processing units and other factors can contribute to the speed of the device. If speed comparisons are the result of comparisons across different (and apparently unknown) graphics cards, the results cannot be directly compared.
Figure 2: The networks shown here need to be described in more detail, both in the caption and in line 194 where the figure is referenced. Also, the figure should be located closer to where it is referenced (it is currently referenced on page 7 but shown on page 5). In fact, Figure 3 is referenced before Figure 2, so the order should likely be switched. Finally, if one network architecture is used throughout the experiments, this should be made clear.
Minor concerns
Line 168: The claim that the proposed method “… exhibits superior performance…” to NLSPN and FCFR-Net should be rephrased to clarify that the superior performance is only by some metrics. NLSPN performs better across 3 error metrics. FCFR_Net performs better on the RMSE and iRMSE metrics.
Line 167: Why is “DeepLiDAR” selected for comparison?
Line 10: “… method outperform multiple…” outperform should be outperforms.
Line 31: “… did not perform well.” – please provide a reference/citation and quantify the performance.
Line 40: “Tang et al[7]. trained a guided…” – remove the period after the [7].
Line 54: “…depth sensors ware widely used,…” – Replace “ware” with “were”.
Line 139: Section header for Section 4 Experiments – “Experiments” should be capitalized.
Line 151: “… no depth value at the top…” – please clarify. Do you mean the top row? Top N rows? What was the original image size? etc
Line 164: You indicate your method is “Net1”, but “Net1” does not appear in the table. Either “(Net1” should be removed or it should be referenced the same way in the table and in the caption. Similarly you abbreviate “SoTA” but never reference this in the table. Which method in the table is the “State of the Art” method that you are referring to?
Line 179: “… other methods at runtime.” I think you mean: “other methods with respect to run time.”
Line 187: I believe this should be reference 20, not reference 19.
Author Response
Dear reviewer,
Thanks for your review and valuable advice. I carefully answered your questions and revised our paper. See below for details.
Major concerns
- I modified this part, see lines 185 to 194 for details. We cannot perform unified testing since FCFR-net does not release the source code. So here we compare NLSPN and GuidedNet.
- We have added a description of the network architecture in the Methods section. (lines 142 to 146)
Minor concerns
1) I modified this part. (line 187)
2) Because DeepLiDAR is trained by additional data. I modified table 1 and made a detailed explanation. See table 1 and line 185 for details.
3) I modified. (line 11)
4) I added a citation. ( line 34)
5) I removed.
6-7) I modified.
8) I modified. (lines 167 to 169)
9) I have revised Table 1. and introduced the network structure. (lines 142 to 146 )
10) I modified.
11) I modified the reference. (line 204)
Reviewer 2 Report
- Line 5 “[…] simply fuse LiDAR information with RGB image information […]”, Line 7 “[…] simply fuse LiDAR information with RGB image information […]”. Repetition. Moreover, the abstract allocate a lot of space to provide an introduction, but allocates 2 lines to introduce a discussion about the proposed idea (“[…] by generating different convolution kernels according to the content and location of feature vectors.”). Please extend the explanation. What exactly is the proposed novelty?
- Please add a space between the previous word and the citation, e.g., “[…] methods[1–3] […]”.
- “[…] Uhrig et al[10]. first proposed […]” please further check the manuscript for typos.
- “[…] how to effectively fuse the information of RGB images and the information of 30 LiDAR.” Please rephrase.
- “Earlier methods directly input RGB images and LiDAR data into a convolutional neural network (CNN) and did not perform well.” Citations?
- “Some recent methods take RGB images and LiDAR data as the input of convolutional neural network separately, and then fuse the features extracted by the convolutional neural network […]”. The procedure is called late feature fusion, please review the literature.
- “Since RGB images and LiDAR data come from different sensors, simple feature concatenation or element-wise addition cannot achieve the best results.” Why not? Please explain. There are many papers that do provide good results, and not only from LiDAR and RGB merging but also from other modalities.
- Last paragraph in Section 1. Why is this important? Why is the proposed method better?
- Please provide an extended introduction for Section 2.
- “At the time of submission, the algorithm ranked first in the official KITTI rankings.” Please rephrase/remove.
- “Therefore, in order to better utilize the image information, we design an adaptive fusion mechanism to fuse information from different sensors.” Please extend. The method would fuse information from different modalities captured by different sensors.
- Figure 1.(a) is blurry. Some white parts are “dirty”.
- Please add some labels to the inputs in Figure 1.(a). The mechanism is not clear. The kernel is applied to both I and S feature maps? How is this different from initial concatenation?
- Add an introduction to section 3. What is the goal of the method, what do it do.
- Please extend section 3.1. “[…] they do not fuse well 104 using the concatenation method.” Why? Based on what observation the authors provide this statement?
- “[…] is reduced to 3 × 3 × k × k × C. our method […]”.
- Section 3.2. It’s not clear what is the novelty. The proposed method is quite simple, and the same results can be achieved using basic layers.
- “[…] Formula 1 is […]”. This is not the correct formulation used in scientific articles. I t can be on in a master thesis, but not in a journal article.
- The conclusion of “Computational efficiency analysis” is not clear. Why is the runtime negligible? Please explain/rephrase the paragraph.
- “4. experiments” ?
- Section 4.
- Figure 2. Please update the caption. “Net 1”? “Net 2”? What are they? Which networks?
- “Our method also exhibits superior performance compared to the NLSPN[29] and 168 FCFR-Net[18] methods.”? The table (Table 1) shows something else. The proposed method provides worse results.
- Table 1 is not useful if the authors don’t discuss the loss function of each method. It MSE is used in the loss function, then the method will perform well when the RMSE/iRMSE metrics are used, and bad for MAE/iMAE. Other methods will perform vice-versa.
- Table 2. What does “200 sample and 500 samples” means? There is a lot of information missing.
- Again the proposed method does not show a clear advantage compare with state-of-the-art.
- Table 3. These are the basic ways of fussing the feature maps, but there are many positions where one can apply it. So it depends also on the position, not only in the fusion method.
- Figure 3 is unclear. Please extend the caption with more information about the colors.
- Figure 4 is extremely blurry. The plots are not clear at all. The legend of Figure 4.(b) has some black dots?
- The discussion section does not provide too many insides.
- The manuscript is missing a conclusion section.
Mai problems:
- There’s not that much novelty proposed.
- The proposed method is not clear.
- The results are not convincing. The comparison with state-of-the-art does not show an improvement.
- The quality of the manuscript must be improved.
Author Response
Dear reviewer,
Thanks for your review and valuable advice. I carefully answered your questions and revised our paper. See below for details.
1) I have revised our abstract. (lines 6 to 13)
2) I modified.
3) I have revised.
4) I have rewritten this sentence. (line 31)
5) I added citations. (line 34)
6) We revised this sentence. (line 35)
7) We have added citations to this passage, and experimental verification has been done in our paper. (line 41 and table 3)
8) We revised our expression and elaborate on the novelty of our approach. (lines 48 to 50)
9) I modified.
10) I removed the sentence.
11) I modified. (lines 97 to 100)
12) I recreated Figure 1(a).
13) I added.
14) I added an additional note. (lines 106 to 108)
15) This view comes from [7]. They also proved it. We added citations to this sentence. (line 118)
16) I modified.
17) We conduct a comparative experiment with the traditionally used basic convolution module. The base convolution is not as good as ours (table 3, Sec. 4.4). Our novelty is to divide the features into different blocks and utilize the attention network to generate different kernel weights for each block. This makes our kernel content dependent. It is dynamic rather than fixed during the inference phase.
18) I modified.
19) I have added an explanation. (line 154)
20-21) I modified.
22) I have added an explanation.( lines 142 to 146 )
23) We modified the expression here. (lines 187 to 194)
24) I don't think this part is a problem, Because many other journal articles are analyzed in the same way[7].
25-26) We have added explanations to the original text.
27-31) I modified.
Main problems:
1) We further illustrate the novelty of our method. (lines 6 to 10 and lines 46 to 50)
2) We further explain our method. (Lines 142 to 146, Lines 152 to 153, Figure 1(a))
3) We revised our experimental results section. (lines 186 to 195)
4) We made changes to the manuscript. (See previous answer for details.)
Reviewer 3 Report
The paper entitled "An adaptive fusion algorithm for depth completion" presents an algorithm that improves on existing algorithms in its application to known data sets.
The work is interesting because of the application it can have in the field by improving current algorithms in different metrics.
The article is well understood, however it has some errors that from my point of view should be corrected.
It is necessary to explain the different acronyms that appear in the text: LiDAR, SLAM...
In the introduction it is necessary to put the different parts of which the article consists.
Figure 4: the captions of the figures and the figures are not uniform.
The discussion section should be expanded commenting on the results obtained and future improvements.
Section 6 should be left out.
Author Response
Dear reviewer,
Thanks for your review and valuable advice. I carefully answered your questions and revised our paper. See below for details.
- We have added explanations for the various acronyms that appear in the text, and we've made changes to the introduction section.(lines 18 to 20 and lines 55 to 57)
- We have modified Figure 4.
- We have revised our Discussion section. See the Conclusion section for details. (lines 242 to 250)
Round 2
Reviewer 1 Report
“An adaptive fusion algorithm for depth completion” by Long Chen and Qing Li
Resubmission feedback:
Thank you for making the updates. After the updates, I still have one major concern and one minor concern.
Major:
Though you made some changes regarding the run time, all reference to run time must either be removed or made exceptionally clear to readers that these timing measurements are made on different graphics cards. This would need to be made clear in the table caption (something like “Run times reported in this table were obtained using different graphics cards and can only provide a rough run time comparison”). Also, in the discussion from lines 182-201, you cannot claim that your method leads all methods in runtime (184-185). Instead, you could state that it is comparable but that direct comparisons cannot be made because each net was run on a different GPU. Instead, you could conclude that your results have roughly the same runtime as many of the other networks evaluated. Similarly, in the abstract, you cannot claim that it outperforms the other networks at runtime because these comparisons are made on different graphics cards.
Minor:
Lines 142 - 146: This could be clarified by instead of stating “We do not introduce too much.”, instead starting the paragraph by stating that the technique developed in this work does not involve new network architecture, so architectures from other networks were used. That way it is clear to the readers that the network architectures tested are not novel. The novelty lies in the adaptive kernel generation.
Author Response
Dear reviewer,
Thanks for your further review. I have carefully revised our paper. See below for details.
Major concerns
I have revised the relevant expressions in the paper. (lines 12 to 13, line 52, line 186, table 1)
Minor concerns
I modified the expression here. (lines 143 to 144)
Reviewer 2 Report
In my opinion the revision of the manuscript was superficial. The main problems mentioned in the first read were not answered.
- There’s not that much novelty proposed.
- The proposed method is not clear.
- The results are not convincing. The comparison with state-of-the-art does not show an improvement. One can note that Tables 1 and 2 demonstrates that the proposed method achieves a worse performance compared with the state-of-the-art methods. Why would one would employ the proposed method instead of state-of-the-art?
- The quality of the manuscript must be improved.
Author Response
Dear reviewer,
Thanks for your further review.
Major problems:
- The novelty of our paper is that we propose an adaptive kernel generation network. It enables our network to generate varying kernels according to the content during the inference phase to fuse different sensor data (lines 46 to 51). This method has not been proposed in the related literature before.
- We use figures, words and formulas to describe our method. Not sure which part of our method you are not clear about, we look forward to your clear instructions.
- Although our method and SoTA have gaps in some metrics, the advantage of our method is that it achieves a good balance between running time and accuracy. Methods with higher accuracy than ours are worse than ours in running time. We believe that accuracy and running time are equally crucial for the development of algorithms.
- Thank you very much for correcting some mistakes in our paper, I really appreciate it. We don't know which part needs to continue to improve. We look forward to your clear instructions.
This manuscript is a resubmission of an earlier submission. The following is a list of the peer review reports and author responses from that submission.
Round 1
Reviewer 1 Report
The manuscript is well written and, after some major revision, could be suitable for publication.
- The manuscript needs to be updated by using another/or big dataset sample sizes for training and testing.
- Clarify the results in the case of training and testing.
- The number of samples for training is much less than for the test, and this affects the results, at least. The ratio is divided into 80/20, 70/30, or 75/25 training and then testing, respectively.
- The performance measures used to mention their results in the abstract and also compare them all with previous research and methods
- Updating references with recent research for 2021 and 2022, and you can use:
DOI 10.7717/peerj-cs.364
Reviewer 2 Report
Journal: Applied Sciences / Journal Not Specified?
Title: An adaptive fusion algorithm for depth completion
Authors: Long Chen and Qing Li
A novel data fusion algorithm has been elaborated for computer visions and suggested for further usage. The algorithm was explained in figures and exhaustive formulas. Two real data sets were used for evaluation of the performance of the suggested algorithm. Four performance merits have been applied. which are linearly not independent.
There are many flaws considering this manuscript, which makes the acceptance dubious.
1) The reproducibility. A scientific paper should be fully reproducible; this trial is far from being that. I doubt that the authors themselves can reproduce their result following solely their descriptions.
2) The validation. A test set validation (KITTI data set) is not sufficient. There are many validation-approaches available: cross-validation, bootstrap, randomization test, etc. There are many variants of cross-validation:
- i) row-wise, pattern-wise (Wold), etc. [R. Bro, K. Kjeldahl, A. K. Smilde, H. A. L. Kiers, Cross-validation of component models: A critical look at current methods, Anal. Bioanal. Chem. 390 (2008) 1241-1251. DOI: 10.1007/s00216-007-1790-1]
- ii) Venetian blinds, contiguous block, etc. [http://wiki.eigenvector.com/index.php?title=Using_Cross-Validation]
iii) repeated double cross-validation [P. Filzmoser, B. Liebmann and K. Varmuza, Repeated double cross validation, J. Chemometrics 23 (2009) 160-171. DOI: 10.1002/cem.1225], etc. Though the robustness has been studied, and the novel method has less error than other two (Figure 4), but 16 other methods exist (Table 2), i.e., the authors have also adapted a questionable comparison.
3) The novelty. Although I do not question that the new algorithm, as such, has some elements of novelty. However, alone the 16 other methods destroy the motivation on a large scale. Moreover, two existing method provide better results: What is the use to develop a worse one?
4) Performance merits: Four linearly NOT independent measures are not sufficient to evaluate the efficiency of the achievement(s). So much the more as a multi-object optimization should have been carried out. A fair method comparison suggests no difference between the known methods (except Revisting[23], which is slightly worse); sum of ranking differences were carried out on the transposed data of Table 2 (and the data were scaled between 0 and 1).
5) Out of scope. The journal publishes miscellaneous material, but I am pretty sure much better publication channels exist; the aimed audience will not understand and apply it.
6) Terminology. The applied terms and expressions are peculiar, e.g., concatenation is almost exclusively used for string concatenation and not for computer vision. Sources: [https://en.wikipedia.org/wiki/Concatenation] and [https://mathworld.wolfram.com/Concatenation.html]
Typos, minor errors
The title is not concrete enough; it does not specify the subject. A title should be brief and concise to the point.
Space usage e.g.: “Huang et al.[2] extended” or “KITTI validation set).We test “– “Huang et al. [2] extended” or “KITTI validation set). We test”, respectively. Such errors can be found in many places of the manuscript.
“RGB images”, “LiDAR sensors”, “ADAM optimizer”, etc. – the abbreviations have not been resolved yet.
Figure 4c: Variance or standard deviation?
RMSE type performance parameters should have units and they depend on the scale.
Citation: Chen, L.; Li, Q. An adaptive fusion algorithm for depth completion. Journal Not Specified 2022, 1, 0. – ???
Two or three flaws enumerated above are enough to reject this manuscript.
March 17 / 2022 referee